# Metabolomics of primary cutaneous melanoma and matched adjacent extratumoral microenvironment

Nicholas J. Taylor[1]*, Irina Gaynanova[2], Steven A. Eschrich[3], Eric A. Welsh[3], Timothy J. Garrett[4], Chris Beecher[5], Ritin Sharma[6¤], John M. Koomen[6], Keiran S. M. Smalley[7,8], Jane L. Messina[8], Peter A. Kanetsky[9]

1 Department of Epidemiology and Biostatistics, Texas A&M University, College Station, Texas, United States of America, 2 Department of Statistics, Texas A&M University, College Station, Texas, United States of America, 3 Department of Biostatistics and Bioinformatics, H. Lee Moffitt Cancer Center & Research Institute, Tampa, Florida, United States of America, 4 Department of Pathology, Immunology and Laboratory Medicine, College of Medicine, University of Florida, Gainesville, Florida, United States of America, 5 IROA Technologies, Chapel Hill, North Carolina, United States of America, 6 Department of Molecular Oncology, H. Lee Moffitt Cancer Center & Research Institute, Tampa, Florida, United States of America, 7 Department of Tumor Biology, H. Lee Moffitt Cancer Center & Research Institute, Tampa, Florida, United States of America, 8 Department of Cutaneous Oncology, H. Lee Moffitt Cancer Center & Research Institute, Tampa, Florida, United States of America, 9 Department of Cancer Epidemiology, H. Lee Moffitt Cancer Center & Research Institute, Tampa, Florida, United States of America

¤ Current address: Collaborative Center for Translational Mass Spectrometry, The Translational Genomics Research Institute, Phoenix, Arizona, United States of America
* ntaylor@tamu.edu

**Data Availability Statement:** Due to Institutional policies related to privacy, the datasets generated and/or analyzed during the current study are not publicly available. Data access requests may be

## Abstract

### Background

Melanoma causes the vast majority of deaths attributable to skin cancer, largely due to its propensity for metastasis. To date, few studies have examined molecular changes between primary cutaneous melanoma and adjacent putatively normal skin. To broaden temporal inferences related to initiation of disease, we performed a metabolomics investigation of primary melanoma and matched extratumoral microenvironment (EM) tissues; and, to make inferences about progressive disease, we also compared unmatched metastatic melanoma tissues to EM tissues.

### Methods

Ultra-high performance liquid chromatography—mass spectrometry-based metabolic profiling was performed on frozen human tissues.

### Results

We observed 824 metabolites as differentially abundant among 33 matched tissue samples, and 1,118 metabolites as differentially abundant between metastatic melanoma (n = 46) and EM (n = 34) after false discovery rate (FDR) adjustment (p<0.01). No significant differences in metabolite abundances were noted comparing primary and metastatic melanoma tissues.

sent to the corresponding author, or the project's IRB administrators at: Advarra 6940 Columbia Gateway Dr. Suite 110 Columbia, MD 21046 Ph. 410-884-2900 cirbi@advarra.com Advarra.com Reference #: CR00189789.

**Funding:** H. Lee Moffitt Cancer Center & Research Institute (Moffitt) Skin Cancer SPORE is funded through NIH grant P50 CA168536. NJT was supported by National Cancer Institute grants: R25 CA147832, T32 CA147832. National Cancer Institutes–Cancer Center Support Grant supports Moffitt Core facilities through P30-CA076292; Moffitt Core facilities are also supported in part by the Moffitt Foundation. Southeastern Center for Integrated Metabolomics (SECIM) is funded by the NIH through U24-DK097209. The funders provided support in the form of salaries for authors [JK, KSMS, PAK], but did not have any additional role in the study design, data collection and analysis, decision to publish, or preparation of the manuscript. The specific roles of these authors are articulated in the 'author contributions' section.

**Competing interests:** I have read the journal's policy and the authors of this manuscript have the following competing interests: Chris Beecher is the Founder and Chief Science Officer of IROA Technologies. Timothy J. Garrett is a member of the Scientific Advisory Board of IROA Technologies. IROA Long Term Reference Standard and Internal Standard were donated by IROA Technologies. Timothy J Garrett's laboratory received no funding from IROA Technologies, and this specific commercial affiliation does not alter the authors' adherence to PLOS ONE policies on sharing data and materials.

## Conclusions

Overall, pathway-based results significantly distinguished melanoma tissues from EM in the metabolism of: ascorbate and aldarate, propanoate, tryptophan, histidine, and pyrimidine. Within pathways, the majority of individual metabolite abundances observed in comparisons of primary melanoma vs. EM and metastatic melanoma vs. EM were directionally consistent. This observed concordance suggests most identified compounds are implicated in the initiation or maintenance of melanoma.

## Background

Melanoma is a neoplasm that evolves through discrete stages distinguished by clinical and pathological features [1, 2]; however, the underlying molecular biology of disease initiation and progression is not well understood. The development and progression of melanoma are influenced by endogenous compounds—metabolites—which combined represent distinct metabolic phenotypes with respect to glucose uptake, glycolytic, and mitochondrial activity [3]. Collectively, metabolites and their interactions constitute the metabolome, a *de facto* representation of cellular physiology that is not subject to epigenetic regulation or post-tran-scriptional/-translational modification, as might be expected when examining genes and proteins, respectively. The metabolome describes metabolic patterns that are most proximal to disease or a phenotype of interest, and may better distinguish key biological mechanisms influencing disease etiology and progression compared to other molecular approaches [4].

To date, most molecular studies of cancer have focused on malignant tissue and have given less consideration to putative histologically-normal tissue surrounding solid tumors, which we hereafter refer to as extratumoral microenvironment (EM). The reasons for this inattention are related to concerns over whether the surrounding tissue is truly normal or, as we posit, represents a subclinical permissive environment for tumor formation on a molecular scale. Previous studies of breast cancer [5, 6], as well as recent pathway-based findings using data from the TCGA [7], are supportive of the latter.

A better temporal understanding of metabolic perturbations influencing melanoma development and progression could further characterize melanoma heterogeneity and identify influential biological pathways, as well as potential therapeutic targets. To identify and evaluate metabolic patterns and differences among temporally distinct tissues, we conducted a metabolomics investigation of frozen primary melanoma tissue and matched EM tissue. And, to better characterize advanced disease, we analyzed a cohort of frozen unmatched metastatic melanoma tissues.

## Methods

### Patient selection

Participants for this study were adults aged 19 to 96 years who were treated for primary cutaneous malignant melanoma at the H. Lee Moffitt Cancer Center and Research Institute (Moffitt)—a National Cancer Institute-designated comprehensive cancer center—and had signed informed written consent for participation in Moffitt's Total Cancer Care Protocol. Study participants' medical records were accessed between July, 2015 and September, 2017 to obtain relevant patient level data. Ethics approval was obtained from Liberty IRB (Liberty IRB Tracking #: 15.06.0013) prior to the start of the study (even after an initial "Non-Human Subjects

Research" determination by the Moffitt Cancer Center's Protocol Support Office: MCC#18133) and all research was performed in accordance with relevant regulations. Melanoma is primarily a disease of Caucasian populations, so eligible cases were limited to patients who self-identified as white. Efforts were made to include participants equitably according to gender, but there were no restrictions on vital status or tumor stage. Eligible cases had existing frozen primary cutaneous melanoma with matching EM tissue available for research purposes. We also selected a different cohort of patients for whom metastatic melanoma tissue (unmatched) was available. Metastases from different organ sites were included to reflect a representative distribution of metastatic lesions observed at the Moffitt Cancer Center and minimize the potential systematic bias that could be introduced by the inclusion of analytes specific to cells other than skin.

## Tissue specimen handling

Primary melanoma and EM specimens as well as metastatic melanoma lesions were procured from patients at the time of surgical excision, and were flash frozen within 30 minutes. H&E-stained frozen sections from each specimen were reviewed by the board-certified study pathologist (JLM) to ensure the presence of tumor at ≥70% cellularity, or in the case of EM tissues, <1% malignant cells. During macrodissection of specimens, efforts were made to exclude tissue regions directly exposed to optimal cutting temperature compound (OCT). OCT is a potentially adulterating compound containing polyvinyl alcohol and polyethylene glycol that is routinely used by pathologists to enable frozen tissue sectioning. To facilitate reproducibility of analyses and ensure adequate tissue availability for future assays, specimens were required to have a minimum mass of 15mg. Tumor specimens were required to exhibit at least 70% tumor cellularity and efforts were made to exclude regions of necrosis and small heterogeneous regions not representative of cells present in the total specimen.

Frozen tissue specimens were stored at -80˚C before processing, which involved biopulverization (Biospec Biopulverizer). After pulverization, each sample was divided into aliquots for metabolomics and future molecular studies, and again stored at -80˚C prior to further analysis.

## IROA and internal standardization

Isotopic Ratio Outlier Analysis (IROA) is a mass spectrometry-based analytic technique that discriminates compounds of biological origin from non-biological artifacts in a two-group or multi-group study [8]. The IROA process was employed in this study and involves the selection of a standardized control cell group that is grown on a medium in which the carbon isotope $^{13}C$ is randomly distributed into all nutrients at 95% abundance (normal or, natural, abundance ~1%). The labeled control cells are then pooled with an equivalent measure of tissue primarily containing the carbon isotope $^{12}C$ (natural abundance ~99%). The pooled samples are then analyzed using liquid chromatography—mass spectrometry (LC-MS) methods. This protocol allows for unambiguous detection of $^{12}C$ and $^{13}C$ monoisotopic peaks for all compounds present in the pooled samples, leading to the accurate differentiation of metabolites found in the standardized control cells versus the tissue specimens. The primary advantage of this method is that only biologically relevant compounds will exhibit incorporation of the $^{13}C$ label in the standard control; thus, only relevant metabolic peaks identified in the tissue specimens will contain matching peaks in the standard control. As a result, significant data reduction is achieved and artifacts that commonly interfere with analysis of mass spectrometry in traditional metabolomics approaches [9] can be readily identified and excluded. This protocol also benefits from unbiased relative quantitation of compounds present in the tissue

compared to the standard control cell line and reduced sample-to-sample variability due to the simultaneous preparation and pooling of standard control and tissue samples.

## IROA standardized control preparation

Yeast (*Saccharomyces* cerevisiae) was selected as the standard control for this study due to the range of metabolites exhibited. Yeast was grown on 5% and 95% $^{13}$C labeled media containing randomized labeled glucose as a single carbon source to enrich the natural abundance of $^{13}$C of all biologically formed metabolites. An extract from 95% $^{13}$C yeast was added as an Internal Standard (IS) to a dried, prepared extract of each homogenized tissue specimen (*e.g.* EM, primary melanoma, or metastatic melanoma) for phenotypic IROA injections, and extracts from both 5% and 95% $^{13}$C yeasts were pooled for the combined IROA injections.

In order to determine the optimal quantities of tissue and labeled control yeast in the pooled analytic sample, we conducted a titration pilot using stand-in meat homogenates. The protein concentration of the pooled meat homogenates was measured. From the stock, 250, 200, 150, 100, 50, and 25μg protein/mL samples were prepared at 50μL. Proteins were precipitated with 400μL 8:1:1 acetonitrile:methanol:acetone and centrifuged at 20,000xg for 10 minutes at 8˚C. The supernatant (375μL) was dried under nitrogen gas at 30˚C, and 10x diluted yeast IROA IS was used to reconstitute samples at a final volume of 40μL. After mass spectra were acquired, 50μg protein/mL was determined to be the optimal protein concentration for sample pre-normalization.

## Metabolite extraction and LC-MS data processing

Pooled samples were pre-normalized at 50μg protein/mL, 25μL prepared as described below, randomized and batched into seven sets that included balanced numbers of samples from each group. Detailed procedures for metabolite extraction and analysis by ultra-high performance liquid chromatography—mass spectrometry (UHPLC-MS) have been previously described [10]. Briefly, the first batch was extracted and immediately placed on LC-MS for analysis. Extraction of subsequent batches was scheduled immediately following the end of the previous batch. Samples (25μL) were extracted using 200μL of 8:1:1 acetonitrile:methanol:acetone and centrifuged at 20,000xg for 10 minutes at 8˚C. 188μL of supernatant was dried down under nitrogen gas at 30˚C and 10x diluted yeast IROA IS was used to reconstitute samples to a final volume of 25μL, thereby assuring that all samples received identical quantities of the IROA IS.

Mass spectra were acquired on a Thermo Scientific Q Exactive orbitrap equipped with a heated electrospray ionization (HESI II) probe in positive ion mode. Full scan mode data were collected in profile mode from *m/z* 70–1000 corresponding to the mass range of most expected metabolites; external calibration (of both mass and mass resolution) was applied before each run to allow for liquid chromatography—high resolution mass spectrometry (LC-HRMS) at 35,000 resolution (*m/z* 200). Chromatographic separation of metabolites was achieved by coupling a Thermo Dionex UltiMate 3000 RS UHPLC system to the Q Exactive Orbitrap. Separation was achieved under gradient elution on an Ace Excel C18-pfp column (100 x 2.1mm, 2μm) at 25˚C. Mobile phase A was 0.1% formic acid in water and B was acetonitrile. The flow rate was 350μL/min starting at 100% A and holding for 3 minutes, followed by an increase in B to 80% over 10 minutes. Mobile phase B was then held constant at 80% for 3 minutes before returning to 100% A for equilibration.

Processing of raw LC-MS data was performed by the Thermo Xcalibur Workstation software (v2.2.44) and Proteowizard's MS Convert (v3.0.5759) was used to centroid and convert Thermo (.raw) files in mzXML data files for further analysis. The mzXML files were analyzed

using ClusterFinder (version 3, IROA Technologies) in order to locate all IROA isotopic peaks and their associated natural abundance companions; results were compared to known metabolite peaks contained within the ClusterFinder library. Quantitation of the natural abundance peak areas and their ratios to the corresponding IROA IS was performed by ClusterFinder.

## Statistical analysis

Principal component analysis (PCA) was performed using the NIPALS [11] algorithm as implemented in the pcaMethods [12] R package, and indicated a significant batch effect after metabolite extraction. The observed batch effect was mitigated by fitting partial least squares (PLS), with the response Y being the batch number (from 1–7). To account for missing data when fitting PLS, we used the R package mixOmics [13]. After visual inspection confirmed that the first component of the PLS analysis contained the batch effect, we computed the resultant residual PLS data matrix with one component which was used for the primary analysis. We repeated PCA using this corrected data to verify no clear separation of batches (Fig 1).

Primary analyses focused on discriminating mean metabolite abundance differences between primary melanoma and matched EM and between metastatic melanoma and unmatched EM. We also explored differences in mean metabolite abundances between primary melanoma and metastatic melanoma. To account for data skewness and an over-representation of zero counts, we conducted both a non-parametric Wilcoxon signed rank test and a parametric t-test; a metabolite was considered noteworthy only if it was found to be significantly differentially abundant between tissue types according to both statistical tests. Paired tests were used for the analysis of matched tissues and unpaired tests were performed for analyses of unmatched tissues. We observed a high level of concordance between the resultant p-values from the two tests when comparing primary melanoma vs. matched EM (Pearson correlation = 0.96) and metastatic melanoma vs. unmatched EM (Pearson correlation = 0.92). To account for multiple comparisons, p-values from each test were adjusted to control the false

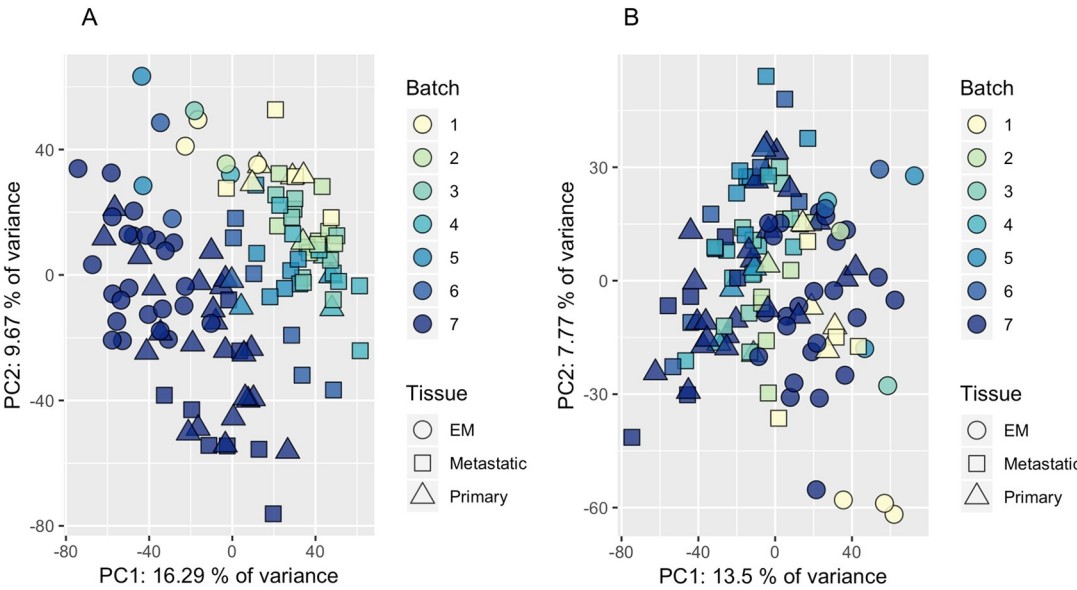

**Fig 1. Identification and mitigation of batch effects.** Results of principal components analysis (PCA) illustrating an observed sample batch effect associated with mass spectrometry (A) and mitigation of that batch effect by fitting partial least squares (PLS) (B).

discovery rate (FDR) using the Benjamini-Yakutieli approach [14] at α = 0.01. Heatmaps illustrating metabolite abundances were generated using the ComplexHeatmap R package [15].

We performed pathway-based analyses comparing primary melanoma vs. EM as well as metastatic melanoma vs. EM, including as input all statistically significant results from both positive and negative ion modes from previous analyses, using Mummichog 1 interfaced through MetaboAnalyst [16–25]. We adopted the Mummichog default p-value cutoff of $1 \times 10^{-05}$ to delineate between significantly enriched and non-significantly enriched metabolites. The KEGG pathways program was then used to identify the most strongly influenced metabolic pathways. We report enrichment factors—the ratio of the number of significant pathway hits to the expected number of compound hits within the pathway—to summarize the influence of individual KEGG pathways.

To further explore functional characteristics of metabolite differences, we performed metabolite set enrichment analysis (MSEA) within MetaboAnalyst. We used the RefMet conversion tool within Metabolomics Workbench [26] to standardize metabolite names prior to MSEA. Enrichment was performed using the Pathway-associated metabolite sets (SMPDB) [27], which consist of 99 metabolite sets from normal pathways. Network analysis was performed in MetaboAnalyst using the metabolite-metabolite interaction network based on associations extracted from STITCH [28].

## Results

### Study samples and patient characteristics

A total of 34 patients with matching frozen primary cutaneous melanoma and EM tissues (mean age at melanoma diagnosis = 64 years), as well as 46 patients with unmatched frozen metastatic melanoma tissue (mean age at melanoma diagnosis = 62 years), were eligible for inclusion in the final analytic cohort. Table 1 describes patient and tissue characteristics of those included in the analytic cohort. Among the 34 patients with matching primary melanoma and EM tissues, 4 patients also had matching metastatic melanoma tissue; these 4 metastatic melanoma tissue specimens were included with the unmatched metastatic melanoma tissue specimens in unmatched analyses only. One patient was excluded from the matched primary melanoma—EM analysis due to primary melanoma sample contamination prior to MS analysis, leaving a total of 33 patients (66 tissue samples) in the final matched cohort (S1 Fig).

### Metabolite detection

In total, 9,599 peaks were detected by UHPLC-MS among the 118 frozen tissue samples included in the study. The final analytic data set excluded 1,540 peaks (16%) due to missing data among at least 50% of the tissue samples.

### Comparison of metabolites in primary melanoma and EM

For the analysis comparing primary melanoma to EM tissues, an additional 552 peaks were excluded from the matched analysis due to failure to detect them among at least 50% of samples contributing to a specific analysis. Of the 7,507 peaks tested among matching tissue samples, 824 (11%) were differentially abundant after FDR adjustment (p<0.01). Approximately 75% of statistically significant metabolites were noted with higher abundance among primary cutaneous melanoma compared to matched EM (S2 Fig). Table 2 lists the top ten percent most significant of the 824 differentially abundant metabolites by mass (*m/z*) and retention time in the matched comparison of primary melanoma vs. EM. A heat map of these peaks is given in Fig 2A, and the paired differences (primary vs. EM) are shown in Fig 2B. Five metabolites

**Table 1. Descriptive characteristics of subjects and tissue specimens contributing to study results.**

| # Subjects | Patients providing primary melanoma tissue | Patients providing metastatic melanoma tissue |
|---|---|---|
| | Total (N = 35) | Total (N = 50) |
| **Gender** | | |
| Men | 24 (*69%*) | 28 (*56%*) |
| Women | 11 (*31%*) | 18 (*36%*) |
| Missing | | 4 (*8%*) |
| **Vital status** | | |
| Alive | 21 (*60%*) | 21 (*42%*) |
| Dead | 14 (*40%*) | 25 (*50%*) |
| Missing | | 4 (*8%*) |
| **Average age at diagnosis (years)**[*] | 64 | 61 |
| **Histology**[*] | | |
| Malignant melanoma, NOS | 15 (*43%*) | |
| Nodular melanoma | 14 (*40%*) | |
| Lentigo maligna melanoma | 1 (*3%*) | |
| Superficial spreading melanoma | 4 (*11%*) | |
| Acral lentignous melanoma | 1 (*3%*) | |
| **Pathologic T Staging (Primaries)** | | |
| pT1 | 1 (*3%*) | |
| pT2 | 0 (*0%*) | |
| pT3 | 8 (*23%*) | |
| pT4 | 15 (*43%*) | |
| pTX | 11 (*31%*) | |
| **Anatomic Site of primary** | | |
| Trunk | 9 (*26%*) | |
| Head/Neck | 12 (*34%*) | |
| Arm/Hand | 7 (*20%*) | |
| Leg/Foot | 7 (*20%*) | |
| Skin, NOS | | |
| **Anatomic Site of metastasis** | | |
| Brain | | 5 (*10%*) |
| Large Bowel | | 1 (*2%*) |
| Lung | | 6 (*12%*) |
| Lymph node | | 13 (*26%*) |
| Salivary gland | | 1 (*2%*) |
| Skin | | 5 (*10%*) |
| Small intestine | | 5 (*10%*) |
| Soft tissue | | 9 (*18%*) |
| Spleen | | 2 (*4%*) |
| Stomach | | 1 (*2%*) |
| Throacic | | 1 (*2%*) |
| Vulva | | 1 (*2%*) |

[*] Data correspond to primary lesion.

**Table 2. Top ten percent of differentially abundant metabolites among primary melanoma and matched EM tissues according to mass (m/z) and retention time.**

| Row* | Ion mode | P-value** | Direction vs. EM[†] | m/z | Retention time (seconds) | Compound Name[‡] | KEGG ID |
|---|---|---|---|---|---|---|---|
| 1 | Negative | 9.10E-06 | 1 | 188.0565 | 2.1 | N-Acetyl-DL-glutamic acid | C00624 |
| 2 | Positive | 5.30E-06 | 1 | 190.0707 | 2.1 | N-Acetyl-DL-glutamic acid | C00624 |
| 3 | Positive | 3.40E-06 | 1 | 113.0346 | 1.3 | Uracil | C00106 |
| 4 | Negative | 2.90E-06 | 1 | 111.0199 | 1.3 | Uracil | C00106 |
| 5 | Positive | 6.60E-06 | 1 | 148.0603 | 0.7 | N-Methyl-D-aspartic acid | C12269 |
| 6 | Positive | 3.00E-06 | 1 | 131.0339 | 0.7 | | |
| 7 | Positive | 6.60E-06 | 1 | 150.0651 | 0.7 | | |
| 8 | Positive | 1.83E-05 | 1 | 148.1092 | 0.6 | | |
| 9 | Positive | 1.78E-05 | 1 | 171.0051 | 0.7 | Dihydroxyacetone phosphate | C00111 |
| 10 | Negative | 6.70E-06 | 1 | 168.9901 | 0.7 | Dihydroxyacetone phosphate | C00111 |
| 11 | Positive | 7.80E-06 | 1 | 210.9975 | 0.7 | | |
| 12 | Positive | 3.40E-06 | 1 | 192.9872 | 0.7 | | |
| 13 | Negative | 1.17E-05 | 1 | 338.9892 | 0.7 | Fructose 1,6-bisphosphate | C00354 |
| 14 | Negative | 9.70E-06 | 1 | 444.0363 | 2.0 | | |
| 15 | Negative | 7.80E-06 | 1 | 372.0843 | 6.3 | | |
| 16 | Negative | 1.87E-05 | 1 | 362.0508 | 3.7 | | |
| 17 | Positive | 3.50E-06 | 1 | 152.0566 | 3.6 | | |
| 18 | Negative | 9.10E-06 | 1 | 346.0558 | 2.7 | | |
| 19 | Positive | 2.90E-06 | 1 | 348.0693 | 2.6 | | |
| 20 | Positive | 1.69E-05 | 1 | 247.0570 | 0.8 | | |
| 21 | Negative | 6.40E-06 | 1 | 245.0431 | 0.8 | | |
| 22 | Negative | 6.60E-06 | 1 | 333.0593 | 0.7 | | |
| 23 | Negative | 3.40E-06 | 1 | 216.9390 | 0.7 | | |
| 24 | Negative | 3.00E-06 | 1 | 126.9440 | 0.7 | | |
| 25 | Negative | 5.00E-06 | 1 | 218.9359 | 0.7 | | |
| 26 | Negative | 2.90E-06 | 1 | 296.8818 | 0.6 | | |
| 27 | Negative | 3.80E-06 | 1 | 346.8808 | 0.6 | | |
| 28 | Negative | 3.00E-06 | 1 | 278.8926 | 0.7 | | |
| 29 | Negative | 6.70E-06 | 1 | 500.8455 | 0.7 | | |
| 30 | Negative | 3.00E-06 | 1 | 432.8575 | 0.6 | | |
| 31 | Negative | 3.00E-06 | 1 | 280.9070 | 0.7 | | |
| 32 | Negative | 2.90E-06 | 1 | 128.9597 | 0.6 | | |
| 33 | Negative | 1.18E-05 | 1 | 445.0531 | 0.8 | | |
| 34 | Positive | 5.40E-06 | 1 | 324.0584 | 0.9 | | |
| 35 | Negative | 5.90E-06 | 1 | 533.1057 | 0.9 | | |
| 36 | Positive | 3.00E-06 | 1 | 245.0603 | 0.9 | | |
| 37 | Negative | 1.47E-05 | 1 | 577.0958 | 0.9 | | |
| 38 | Positive | 1.15E-05 | 1 | 213.0163 | 1.2 | | |
| 39 | Positive | 1.18E-05 | 1 | 123.5941 | 0.9 | | |
| 40 | Positive | 6.40E-06 | 1 | 186.0758 | 3.0 | | |
| 41 | Positive | 6.60E-06 | 1 | 292.1018 | 1.2 | | |
| 42 | Positive | 1.18E-05 | 1 | 186.0761 | 0.7 | | |
| 43 | Positive | 1.13E-05 | 1 | 130.1227 | 1.7 | | |
| 44 | Negative | 1.60E-05 | 1 | 98.9737 | 0.7 | | |
| 45 | Negative | 2.90E-06 | 1 | 96.9693 | 0.7 | | |
| 46 | Negative | 5.00E-06 | 1 | 186.9649 | 0.7 | | |
| 47 | Positive | 5.40E-06 | 1 | 267.0562 | 0.8 | | |

(*Continued*)

**Table 2.** (Continued)

| Row[*] | Ion mode | P-value[**] | Direction vs. EM[†] | m/z | Retention time (seconds) | Compound Name[‡] | KEGG ID |
|---|---|---|---|---|---|---|---|
| 48 | Negative | 1.72E-05 | 1 | 451.1265 | 0.8 | | |
| 49 | Positive | 6.60E-06 | 1 | 177.9573 | 0.7 | | |
| 50 | Positive | 3.50E-06 | 1 | 193.9296 | 0.7 | | |
| 51 | Positive | 9.80E-06 | 1 | 195.9679 | 0.7 | | |
| 52 | Positive | 1.45E-05 | 1 | 154.9414 | 0.7 | | |
| 53 | Positive | 3.00E-06 | -1 | 218.9191 | 0.6 | | |
| 54 | Negative | 2.90E-06 | -1 | 312.9429 | 0.6 | | |
| 55 | Positive | 7.80E-06 | -1 | 349.2111 | 0.7 | | |
| 56 | Negative | 7.70E-06 | -1 | 315.9351 | 0.6 | | |
| 57 | Negative | 6.70E-06 | -1 | 358.9227 | 0.6 | | |
| 58 | Negative | 1.60E-05 | -1 | 468.8964 | 0.6 | | |
| 59 | Negative | 5.40E-06 | -1 | 536.8855 | 0.6 | | |
| 60 | Negative | 5.80E-06 | -1 | 248.9604 | 0.6 | | |
| 61 | Negative | 3.80E-06 | -1 | 316.9480 | 0.6 | | |
| 62 | Negative | 6.40E-06 | -1 | 446.9031 | 0.6 | | |
| 63 | Negative | 2.90E-06 | -1 | 314.9326 | 0.6 | | |
| 64 | Negative | 2.90E-06 | -1 | 246.9449 | 0.6 | | |
| 65 | Negative | 3.00E-06 | -1 | 380.9042 | 0.6 | | |
| 66 | Positive | 3.50E-06 | -1 | 158.9615 | 0.9 | | |
| 67 | Positive | 3.40E-06 | -1 | 226.9510 | 0.6 | | |
| 68 | Positive | 5.00E-06 | -1 | 90.9766 | 0.7 | | |
| 69 | Positive | 5.80E-06 | -1 | 156.9483 | 0.6 | | |
| 70 | Positive | 5.80E-06 | -1 | 82.0264 | 0.6 | | |
| 71 | Positive | 6.60E-06 | -1 | 288.9207 | 0.6 | | |
| 72 | Negative | 1.88E-05 | -1 | 94.9247 | 0.7 | | |
| 73 | Negative | 7.00E-06 | -1 | 92.9278 | 0.7 | | |
| 74 | Negative | 7.80E-06 | -1 | 96.9218 | 0.7 | | |
| 75 | Negative | 1.66E-05 | -1 | 154.8986 | 0.8 | | |
| 76 | Negative | 6.40E-06 | -1 | 181.9661 | 0.7 | | |
| 77 | Negative | 3.00E-06 | -1 | 113.0243 | 0.7 | | |
| 78 | Negative | 2.90E-06 | -1 | 89.0239 | 0.7 | | |
| 79 | Negative | 7.10E-06 | -1 | 119.0353 | 0.7 | | |
| 80 | Negative | 5.80E-06 | -1 | 143.0354 | 0.7 | | |
| 81 | Negative | 2.90E-06 | -1 | 217.0299 | 0.7 | Glucose/Fructose | |
| 82 | Negative | 2.90E-06 | -1 | 215.0327 | 0.7 | Glucose/Fructose | |

[*]Corresponds to row in Fig 2.

[**]Minimum FDR adjusted P-value among Wilcoxon signed rank test and T-test.

[†]1 = metabolite is more abundant in tumor relative to EM; -1 = metabolite is less abundant in tumor relative to EM.

[‡] Compound identification according to IROA ClusterFinder library.

falling in the top ten percent of statistically significant results had verified KEGG Compound IDs. Three of these five metabolites had significantly higher abundances among primary melanoma vs. EM in both positive and negative ion modes: N-Acetyl-DL-glutamic acid (KEGG ID: C00624); Uracil (KEGG ID: C00106); and Dihydroxyacetone phosphate (KEGG ID: C00111) (S3 Fig). Additionally, N-Methyl-D-aspartic acid (KEGG ID: C12269) and Fructose 1,6-bisphosphate (KEGG ID: C00354) had significantly higher abundances in primary

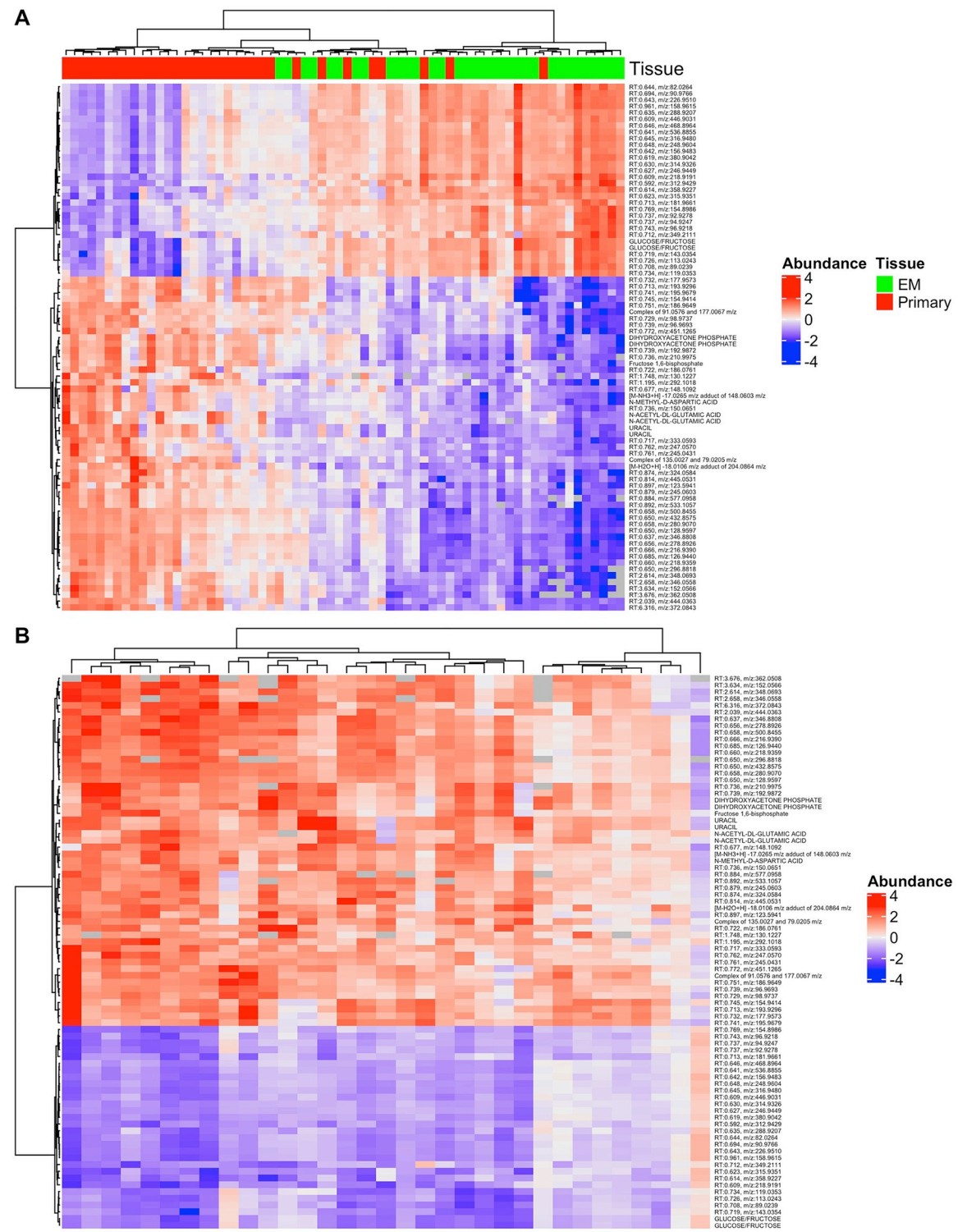

**Fig 2. Metabolite abundance profiles differ significantly between primary melanoma and matched EM tissues.** (A) Heatmap of the top 10% of significant metabolites (n = 824). Tissue type (EM: Green or primary melanoma: Red) is noted in the top color bar. Unidentified metabolites are annotated with retention time (RT) and mass (*m/z*). (B) Heatmap of the metabolite differences between matched primary melanoma and EM tissues from the same patient for the top 10% of significant metabolites.

melanoma vs. EM in positive ion mode and negative ion mode, respectively (S3 Fig). Supplemental bar graphs illustrate all metabolites that were identified as significantly differentially abundant between primary melanoma tissue and paired EM (S4 Fig).

## Comparison of metabolites in metastatic melanoma and EM

The analysis comparing metastatic melanoma to EM tissues was performed after an additional 294 peaks were excluded due to failure to detect them among at least 50% of EM samples. Among the 7,765 peaks subsequently tested in the comparison of metastatic melanoma and EM tissues, 1,118 (14%) were differentially abundant after FDR adjustment (p<0.01). Approximately 67% of statistically significant metabolites were observed at higher abundances among metastatic melanoma compared to EM (S2 Fig). Of the 1,118 differentially abundant metabolites identified, the top ten percent most significant are presented in Table 3 according to mass (*m/z*) and retention time and are mapped in Fig 3. Four metabolites falling in the top ten percent of statistically significant results had verified KEGG IDs based on *m/z* and retention times; one of four metabolites was significantly higher in abundance among metastatic melanoma vs. EM in both positive and negative ion modes: Uracil (KEGG ID: C00106). The remaining three, Cytidine 5'-diphosphocholine (KEGG ID: C00307), Cytosine (KEGG ID: C00380), and N-Methyl-D-aspartic acid (KEGG ID: C12269) were higher in abundance among metastatic melanoma compared to EM in positive ion mode (S5 Fig). Supplemental box plots illustrate all metabolites that were identified as significantly differentially abundant between EM and unmatched metastatic melanoma (S6 Fig). A comparison of metabolite abundances found in primary melanomas and unmatched metastatic melanomas yielded no statistically significant differences at the adjusted p<0.01 threshold (S7 Fig).

## KEGG pathway-based analysis of metabolites

KEGG pathway-based analysis of metabolites distinguishing primary melanoma and matched EM showed significant influence of compounds involved in the metabolism of: ascorbate and aldarate (Enrichment Factor = 1.67, p = 0.003); propanoate (Enrichment Factor = 0.96, p = 0.04); and tryptophan (Enrichment Factor = 0.65, p = 0.04). Metabolites involved in vitamin B6 metabolism (Enrichment Factor = 1.21, p = 0.06) and pyruvate metabolism (Enrichment Factor = 1.37, p = 0.08) were identified as strongly enriched, but were not statistically significant (Table 4).

KEGG pathway-based analysis including metabolites identified in the unmatched analysis of EM and metastatic melanoma showed pronounced influence of compounds involved in ascorbate and aldarate metabolism (Enrichment Factor = 1.33, p = 0.002). Additionally, pyrimidine (Enrichment Factor = 1.03, p = 0.01) and histidine (Enrichment Factor = 0.81, p = 0.03) metabolism were significantly influential in distinguishing metastatic melanoma from EM. Pentose phosphate metabolism (Enrichment Factor = 1.46, p = 0.05) was notably enriched, but marginally statistically significant (Table 4).

## MSEA pathway-based analysis of metabolites

We conducted MSEA for metabolites with statistically significantly higher abundance (or lower abundance) in primary (or metastatic) melanoma compared to EM; and for each condition, the MSEA of pathway-associated metabolite sets—based on the small molecule pathway database (SMPDB) [27]—was considered. A network of metabolite-metabolite interactions was plotted for each condition, with nodes representing metabolites and edges indicating a relationship between metabolites based on STITCH databases [28]. L-glutamic acid and pyruvic acid exhibited the most interactions when we considered significantly overabundant

**Table 3. Top ten percent of differentially abundant metabolites among metastatic melanoma and EM tissues according to mass (m/z) and retention time.**

| Row* | Ion mode | P-value** | Direction vs. EM† | m/z | Retention time (seconds) | Compound Name | KEGG ID |
|---|---|---|---|---|---|---|---|
| 1 | Negative | 4.28E-08 | 1 | 218.0671 | 3.2 | | |
| 2 | Positive | 4.00E-11 | 1 | 230.1129 | 1.5 | | |
| 3 | Positive | 2.59E-08 | 1 | 264.0341 | 0.9 | | |
| 4 | Positive | 5.80E-10 | 1 | 272.0199 | 0.9 | | |
| 5 | Negative | 1.15E-08 | 1 | 533.1057 | 0.9 | | |
| 6 | Positive | 5.89E-09 | 1 | 245.0603 | 0.9 | | |
| 7 | Positive | 1.98E-08 | 1 | 489.1135 | 0.9 | Cytidine 5'-diphosphocholine | C00307 |
| 8 | Negative | 3.61E-09 | 1 | 577.0958 | 0.9 | | |
| 9 | Positive | 1.50E-10 | 1 | 112.0505 | 0.8 | Cytosine | C00380 |
| 10 | Positive | 9.60E-09 | 1 | 204.0864 | 3.0 | | |
| 11 | Positive | 1.19E-09 | 1 | 186.0758 | 3.0 | | |
| 12 | Negative | 5.66E-09 | 1 | 606.0745 | 3.0 | | |
| 13 | Positive | 4.91E-09 | 1 | 204.0863 | 3.1 | | |
| 14 | Negative | 4.80E-08 | 1 | 606.0745 | 3.2 | | |
| 15 | Positive | 1.41E-09 | 1 | 204.0863 | 3.3 | | |
| 16 | Negative | 3.80E-10 | 1 | 333.0593 | 0.7 | | |
| 17 | Positive | 1.10E-10 | 1 | 335.0726 | 0.7 | | |
| 18 | Negative | 4.40E-10 | 1 | 245.0431 | 0.8 | | |
| 19 | Positive | 1.50E-10 | 1 | 247.0570 | 0.8 | | |
| 20 | Positive | 8.91E-08 | 1 | 148.0603 | 0.7 | N-Methyl-D-aspartic acid | C12269 |
| 21 | Positive | 3.63E-08 | 1 | 131.0339 | 0.7 | | |
| 22 | Positive | 5.00E-10 | 1 | 150.0651 | 0.7 | | |
| 23 | Negative | 1.98E-08 | 1 | 228.0639 | 0.7 | | |
| 24 | Negative | 1.39E-08 | 1 | 272.0540 | 0.7 | | |
| 25 | Positive | 1.10E-07 | 1 | 706.0822 | 7.8 | | |
| 26 | Positive | 1.10E-07 | 1 | 706.2252 | 7.8 | | |
| 27 | Negative | 9.21E-08 | 1 | 160.0823 | 0.7 | | |
| 28 | Negative | 4.40E-10 | 1 | 126.9440 | 0.7 | | |
| 29 | Negative | 1.50E-10 | 1 | 216.9390 | 0.7 | | |
| 30 | Negative | 6.00E-11 | 1 | 218.9359 | 0.6 | | |
| 31 | Negative | 6.00E-11 | 1 | 128.9597 | 0.6 | | |
| 32 | Negative | 6.00E-11 | 1 | 296.8818 | 0.6 | | |
| 33 | Negative | 1.65E-08 | 1 | 346.8808 | 0.6 | | |
| 34 | Negative | 1.27E-09 | 1 | 278.8926 | 0.6 | | |
| 35 | Negative | 1.36E-09 | 1 | 432.8575 | 0.6 | | |
| 36 | Negative | 2.40E-10 | 1 | 280.9070 | 0.6 | | |
| 37 | Negative | 8.20E-08 | 1 | 500.8455 | 0.6 | | |
| 38 | Positive | 6.00E-10 | 1 | 156.9981 | 1.3 | | |
| 39 | Positive | 1.10E-10 | 1 | 123.5234 | 1.3 | | |
| 40 | Negative | 3.00E-11 | 1 | 173.0203 | 1.3 | | |
| 41 | Positive | 1.00E-11 | 1 | 113.0346 | 1.3 | Uracil | C00106 |
| 42 | Negative | 1.00E-11 | 1 | 111.0199 | 1.3 | Uracil | C00106 |
| 43 | Positive | 4.40E-10 | 1 | 114.5181 | 1.3 | | |
| 44 | Positive | 9.00E-11 | 1 | 214.9969 | 1.3 | | |
| 45 | Positive | 3.18E-08 | 1 | 126.0268 | 1.4 | | |
| 46 | Positive | 1.50E-10 | 1 | 230.9694 | 1.3 | | |
| 47 | Positive | 9.00E-11 | 1 | 212.9581 | 1.3 | | |

*(Continued)*

**Table 3.** (Continued)

| Row* | Ion mode | P-value** | Direction vs. EM† | m/z | Retention time (seconds) | Compound Name | KEGG ID |
|------|----------|-----------|-------------------|-----|--------------------------|---------------|---------|
| 48 | Positive | 1.98E-08 | 1 | 268.1033 | 6.4 | | |
| 49 | Positive | 1.59E-09 | 1 | 290.0852 | 6.4 | | |
| 50 | Negative | 6.26E-09 | 1 | 266.0898 | 6.4 | | |
| 51 | Negative | 8.20E-10 | 1 | 306.0498 | 1.2 | | |
| 52 | Positive | 2.80E-10 | 1 | 395.4083 | 7.2 | | |
| 53 | Positive | 2.59E-08 | 1 | 325.0422 | 2.0 | | |
| 54 | Positive | 2.28E-09 | 1 | 213.0159 | 2.0 | | |
| 55 | Negative | 8.15E-09 | 1 | 323.0286 | 2.0 | | |
| 56 | Positive | 8.25E-09 | 1 | 347.0240 | 2.0 | | |
| 57 | Negative | 8.25E-09 | 1 | 435.0059 | 2.0 | | |
| 58 | Negative | 1.85E-08 | 1 | 346.0558 | 2.6 | | |
| 59 | Positive | 1.16E-08 | 1 | 348.0693 | 2.6 | | |
| 60 | Positive | 1.20E-07 | 1 | 152.0566 | 3.6 | | |
| 61 | Positive | 2.55E-08 | 1 | 357.0994 | 5.7 | | |
| 62 | Positive | 1.67E-08 | 1 | 358.1028 | 5.7 | | |
| 63 | Negative | 8.71E-09 | 1 | 356.0893 | 5.7 | | |
| 64 | Positive | 1.90E-10 | 1 | 374.0977 | 6.3 | | |
| 65 | Negative | 1.10E-10 | 1 | 372.0843 | 6.3 | | |
| 66 | Negative | 8.30E-08 | 1 | 332.0587 | 2.5 | | |
| 67 | Negative | 7.58E-08 | 1 | 105.0016 | 15.8 | | |
| 68 | Positive | 7.06E-08 | 1 | 334.0722 | 2.0 | | |
| 69 | Positive | 2.69E-08 | 1 | 356.0540 | 2.0 | | |
| 70 | Negative | 1.61E-08 | 1 | 444.0363 | 2.0 | | |
| 71 | Negative | 1.06E-08 | 1 | 332.0589 | 2.0 | | |
| 72 | Positive | 2.69E-08 | 1 | 218.0323 | 2.0 | | |
| 73 | Negative | 1.06E-08 | -1 | 340.9375 | 0.6 | | |
| 74 | Negative | 3.70E-10 | -1 | 312.9429 | 0.6 | | |
| 75 | Negative | 1.59E-09 | -1 | 181.9661 | 0.7 | | |
| 76 | Positive | 4.32E-08 | -1 | 216.9216 | 0.6 | | |
| 77 | Positive | 3.20E-10 | -1 | 218.9191 | 0.6 | | |
| 78 | Positive | 1.14E-07 | -1 | 150.0139 | 0.7 | | |
| 79 | Negative | 5.30E-10 | -1 | 315.9351 | 0.6 | | |
| 80 | Negative | 3.70E-10 | -1 | 358.9227 | 0.6 | | |
| 81 | Positive | 1.50E-10 | -1 | 226.9510 | 0.6 | | |
| 82 | Positive | 1.40E-10 | -1 | 90.9766 | 0.7 | | |
| 83 | Positive | 9.00E-11 | -1 | 158.9615 | 1.0 | | |
| 84 | Positive | 7.40E-10 | -1 | 82.0264 | 0.6 | | |
| 85 | Positive | 1.21E-09 | -1 | 288.9207 | 0.6 | | |
| 86 | Negative | 3.80E-10 | -1 | 446.9031 | 0.6 | | |
| 87 | Positive | 2.54E-08 | -1 | 236.9065 | 0.6 | | |
| 88 | Negative | 2.04E-09 | -1 | 246.9449 | 0.6 | | |
| 89 | Negative | 1.59E-09 | -1 | 314.9326 | 0.6 | | |
| 90 | Negative | 1.00E-10 | -1 | 380.9042 | 0.6 | | |
| 91 | Negative | 1.67E-08 | -1 | 468.8964 | 0.6 | | |
| 92 | Negative | 1.58E-09 | -1 | 536.8855 | 0.6 | | |
| 93 | Negative | 2.90E-09 | -1 | 248.9604 | 0.6 | | |
| 94 | Negative | 4.60E-10 | -1 | 316.9480 | 0.6 | | |

(*Continued*)

**Table 3.**  (Continued)

| Row* | Ion mode | P-value** | Direction vs. EM† | m/z | Retention time (seconds) | Compound Name | KEGG ID |
|------|----------|-----------|-------------------|-----|--------------------------|---------------|---------|
| 95 | Positive | 7.40E-10 | -1 | 156.9483 | 0.6 | | |
| 96 | Negative | 1.31E-08 | -1 | 144.8697 | 0.8 | | |
| 97 | Negative | 1.50E-10 | -1 | 146.8667 | 0.8 | | |
| 98 | Negative | 6.80E-10 | -1 | 154.8809 | 0.8 | | |
| 99 | Negative | 1.11E-08 | -1 | 94.9247 | 0.7 | | |
| 100 | Negative | 4.91E-09 | -1 | 92.9278 | 0.7 | | |
| 101 | Negative | 1.31E-08 | -1 | 154.8986 | 0.8 | | |
| 102 | Negative | 7.12E-08 | -1 | 96.9218 | 0.7 | | |
| 103 | Positive | 1.10E-07 | -1 | 241.1588 | 2.3 | | |
| 104 | Positive | 2.63E-09 | -1 | 256.1775 | 6.3 | | |
| 105 | Positive | 2.58E-08 | -1 | 257.1614 | 1.8 | | |
| 106 | Positive | 1.10E-07 | -1 | 200.0912 | 4.7 | | |
| 107 | Negative | 4.28E-08 | -1 | 113.0243 | 0.7 | | |
| 108 | Negative | 2.82E-08 | -1 | 143.0354 | 0.7 | | |
| 109 | Negative | 8.29E-08 | -1 | 89.0239 | 0.7 | | |
| 110 | Negative | 3.61E-09 | -1 | 217.0299 | 0.7 | Glucose/Fructose | |
| 111 | Negative | 3.30E-09 | -1 | 215.0327 | 0.7 | Glucose/Fructose | |
| 112 | Positive | 3.16E-08 | -1 | 180.0865 | 0.7 | | |

*Corresponds to row in Fig 3.

**Minimum FDR adjusted P-value among Wilcoxon signed rank test and T-test.

†1 = metabolite is more abundant in tumor relative to EM; -1 = metabolite is less abundant in tumor relative to EM.

‡ Compound identification according to IROA ClusterFinder library.

metabolites among primary melanoma compared to EM (S8 Fig), while D-glucose and D-galactose exhibited the most interactions when we considered metabolites that were significantly less abundant among primary melanoma compared to EM (S9 Fig).

For those metabolites that were overabundant among metastatic melanoma vs. EM, we again noted that L-glutamic acid and pyruvic acid exhibited the most interactions in the MSEA generated metabolite network (S10 Fig). Similarly to results from our comparison of primary melanoma and EM, D-glucose and D-galactose exhibited the most interactions when we considered metabolites that were significantly less abundant among metastatic melanoma compared to EM (S11 Fig).

## Discussion

To date, metabolomics investigations of frozen human tissue specimens in the context of cancer have been limited; especially scant are those involving non-malignant tissue. To our knowledge, this study represents the largest examination of extratumoral and matched primary cutaneous melanoma tissues as well as a cohort of unmatched metastatic melanomas.

Our evaluation showed significantly differentially abundant metabolites comparing either primary melanoma or metastatic melanoma to EM, and that 75% and 68% of differences, respectively, represented higher levels of metabolite in the melanoma tissue. In contrast, comparison of metabolite abundances between primary and metastatic melanoma yielded no statistically significant differences. In a study of melanoma cell lines, Yu *et al.* reported 12 metabolites different between primary melanoma and descendent metastatic melanoma [29]. This study applied a less conservative significance level ($\alpha = 0.05$) whereas our approach was

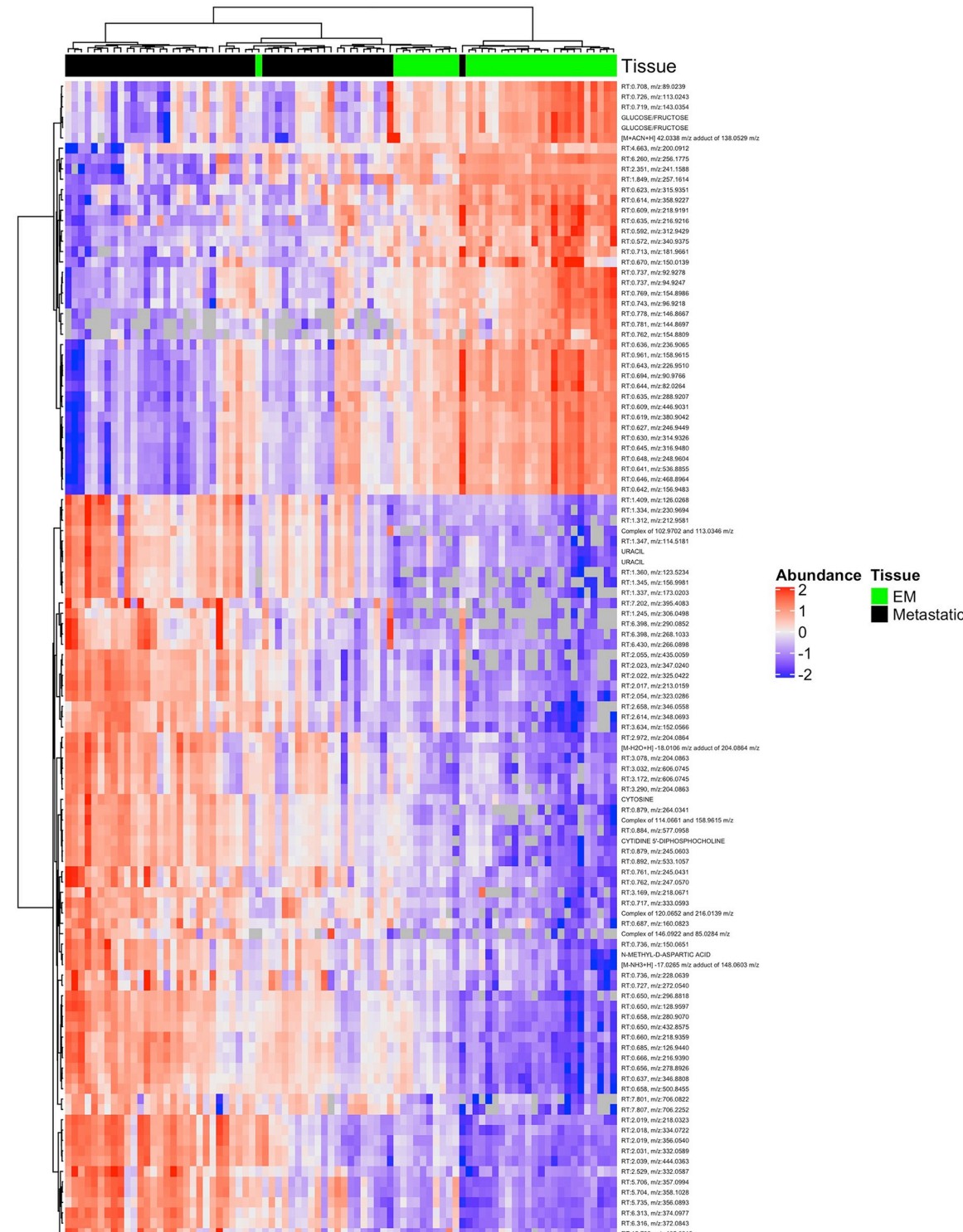

**Fig 3. Metabolite abundance profiles differ significantly between metastatic melanoma and unmatched EM tissues.** Heatmap of the top 10% of significant metabolites (n = 1,118). Tissue type (EM:green or metastatic melanoma: Black) is noted in the top color bar.

**Table 4. Top KEGG biological pathways distinguishing primary melanoma from EM and metastatic melanoma from EM.**

| Primary melanoma vs. EM | | | | | Metastatic melanoma vs. EM | | | | |
|---|---|---|---|---|---|---|---|---|---|
| Pathway name | Total metabolites* | Hits (all/sig)** | Expected† | Enrichment factor‡ | P-value | Total metabolites* | Hits (all/sig)** | Expected† | Enrichment factor‡ | P-value |
| Ascorbate and aldarate | 45 | 32/29 | 17.41 | 1.67 | 0.003 | 45 | 32/32 | 24.1 | 1.33 | 0.002 |
| Tryptophan | 79 | 23/20 | 30.56 | 0.65 | 0.04 | 79 | 26/23 | 42.31 | 0.54 | 0.33 |
| Propanoate | 35 | 14/13 | 13.54 | 0.96 | 0.04 | 35 | 15/12 | 18.74 | 0.64 | 0.76 |
| Vitamin B6 | 32 | 17/15 | 12.38 | 1.21 | 0.06 | 32 | 19/17 | 17.14 | 0.99 | 0.35 |
| Pyruvate | 32 | 20/17 | 12.38 | 1.37 | 0.08 | 32 | 20/14 | 17.14 | 0.82 | 0.96 |
| Pyrimidine | 60 | 35/24 | 23.21 | 1.03 | 0.57 | 60 | 34/33 | 32.13 | 1.03 | 0.01 |
| Histidine | 44 | 17/12 | 17.02 | 0.71 | 0.54 | 44 | 19/19 | 23.56 | 0.81 | 0.03 |
| Pentose phosphate | 32 | 24/14 | 12.38 | 1.13 | 0.90 | 32 | 26/25 | 17.14 | 1.46 | 0.05 |

* Total number of KEGG metabolites considered by Mummichog/Metaboanalyst according to pathway.

** Total number of hits per pathway and total number of statistically significant hits per pathway ($p < 1 \times 10^{-05}$).

† Expected number of hits.

‡ The ratio of the number of significant pathway hits to the expected number of compound hits within the pathway.

deliberately conservative to identify differences with high confidence focused on malignant vs. EM conditions. Our data suggest that primary vs. metastatic differences may be more subtle than differences observed between malignant tissues and EM, and future studies are needed to fully characterize these differences.

Overall, pathway-based analysis yielded three significantly influential pathways in distinguishing primary melanoma from EM (ascorbate and aldarate metabolism, propanoate metabolism, and tryptophan metabolism) and three significantly influential pathways in distinguishing metastatic melanoma from EM (ascorbate and aldarate metabolism, pyrimidine metabolism and histidine metabolism) that may reflect important foci of carcinogenesis and cancer progression, respectively. Within these pathways, the majority of individual metabolite abundances observed in comparisons of primary melanoma vs. EM and metastatic melanoma vs. EM were directionally consistent; that is, the majority of compounds that were observed to be higher in abundance (or less abundant) in primary melanoma compared to EM were also higher in abundance (or less abundant) in metastatic melanoma compared to EM. This observed concordance suggests most identified compounds are implicated in the initiation or maintenance of melanoma, as opposed to progression of disease. Nevertheless, we observed several key metabolites within pathways that were in opposing directions according to primary melanoma vs. EM or metastatic melanoma vs. EM. These discordant observations are suggestive of metabolites that may be influential in disease progression.

Ascorbate and aldarate metabolism showed notable differences between both malignant tissue types and EM. Interestingly, within this pathway, arabinose was observed at lower abundances in primary melanoma compared to EM, whereas it was at higher levels in metastatic melanoma compared to EM. An overabundance of arabinose was reported in colorectal tumors compared to matched normal mucosa, where the majority of malignant tissues sampled (56%) represented metastatic disease [30]. Similarly, in a study comparing serum metabolite concentrations between advanced pancreatic cancer patients (79% stage III/IV) and healthy volunteers, arabinose was found at nearly 2-fold higher concentrations among pancreatic cancer patients [31]. In contrast, a study comparing relatively early staged esophageal carcinomas (60% stage I/II; 70% non-metastatic) to matched normal mucosae reported a lower abundance of arabinose among the tumor tissue [32]. These results may indicate that

arabinose and related metabolic pathways should be more thoroughly investigated. In addition, arabinose has been the target of bacteria-based cytotoxic therapies in murine colorectal tumors involving *Salmonella typhimurium*. The attenuated strain is engineered to exclude the *ara* operon that is responsible for metabolizing arabinose to D-xylulose-5-phosphate. When affected mice were treated with the operon-deleted strain, arabinose accumulation resulted, facilitating expression of the cytotoxic protein cytolysin A in the tumors [33]. In conjunction with arabinose supplementation, such therapies may show promise among earlier staged melanomas, as well as in the treatment of metastatic melanomas that accumulate high levels of arabinose. It should be noted that compounds involved in the KEGG annotated ascorbate and aldarate metabolic pathway are involved in several other pathways, leading some to suggest the pathway is redundant and should not be considered as an independent pathway [34].

We noted a significant influence of compounds involved in propanoate metabolism on distinguishing primary melanoma from EM. Propanoate was notably reduced among primary melanoma compared to EM, whereas it was at significantly higher levels among metastatic melanoma compared to EM. Propanoate has been shown to inhibit tumor cell proliferation and promote apoptosis in different cancer cell types [35–37]; one possible explanation for our observation may be related to reduced expression of free fatty acid receptor 2 (FFAR2, also known as GPR43)—a receptor for which propanoate is the most potent ligand [38]—leading to an unproductive pool of propanoate. Loss of FFAR2 has been shown to promote colon cancer and leukemia in murine models [39–41], and expression levels are relatively diminished among TCGA melanoma tissues vs. several other cancers according to the Human Protein Atlas [42]. Loss of FFAR2 (or obstruction of ligand binding) may be a significant marker of melanoma progression, and is already being examined as a therapeutic target for metabolic and inflammatory diseases [43]. Surprisingly, in comparisons of both primary melanoma and metastatic melanoma vs. EM, we observed a reduced abundance of L-lactate. Lactate is involved in propanoate metabolism as a contributor to the pool of propionyl-CoA, which is converted to succinyl-CoA and enters the Krebs cycle. It is also an important factor in glycolysis. Increased production of lactate via glycolysis is a hallmark of cancer in general, as the vast majority of human cancers exhibit overexpression of glycolytic genes [44, 45]. Melanoma cells are known to exhibit the Warburg effect in cell line studies, but have also demonstrated functional oxidative phosphorylation, even under hypoxic conditions [46]. However, in the context of tumor microenvironment, newer research suggests metabolic heterogeneity across tumors [47] that may be attributed to a preference for oxidative phosphorylation—a so-called "reverse Warburg effect" [48–51]. This theory involves aberrant induction of aerobic glycolysis in adjacent cells and utilization of the products for mitochondrial oxidative phosphorylation in tumor cells [52–55]. The proposed mechanism involves tumor cell stimulation of oxidative stress among adjacent fibroblasts, leading to downregulation of caveolin-1 (CAV1)—a plasma membrane protein associated with increased mitochondrial activity [55, 56]. Downregulated CAV1 promotes glycolysis in adjacent cells leading to increased tumor growth via the reverse Warburg effect [57]. Loss of stromal CAV1 has been associated with poorer prognosis among metastatic melanoma patients [58], and clinical studies of squamous cell carcinoma have demonstrated reconstitution of CAV1 and increased tumor cell apoptosis in patients treated with Metformin [59]—a drug known to inhibit mitochondrial oxidative phosphorylation.

Endogenous molecules related to tryptophan metabolism could also distinguish primary melanoma from EM. Notably, N-acetylserotonin was less abundant among primary melanoma vs. EM and more abundant in metastatic melanoma vs. EM, whereas melatonin levels were not significantly different between melanoma tissues and EM. Melanoma cells can possess high and low affinity binding sites for melatonin, which is known to inhibit cell proliferation [60]. Souza *et al*. demonstrated equally potent antiproliferative effects for N-acetylserotonin

and melatonin [60]; however, N-acetylserotonin only binds low affinity receptors whereas melatonin binds both high and low affinity sites. As N-acetylserotonin is a precursor of melatonin, and both ligands have similar binding affinities, our results may suggest inhibited binding of an extant, but limited, pool of melatonin.

Pyrimidine and histidine metabolic pathways accounted for some of the differences between the metabolomes of metastatic melanoma and EM. With respect to pyrimidine metabolism, cytidine was notably increased among metastatic melanoma relative to EM, but reduced among primary melanoma vs. EM. One explanation may be reduced expression of cyditine deaminase (CDA), which is responsible for the irreversible conversion of cytidine to uridine. CDA has recently been reported as downregulated in approximately 60% of cancer cells and tissues, making it a promising therapeutic target [61]. The histidine metabolic pathway also exhibited individual metabolite differences when comparing metastatic melanoma vs. EM to primary melanoma vs. EM results. Notably, urocanic acid (UCA) was detected at higher levels among primary melanoma vs. EM and was less abundant in metastatic melanoma vs. EM, whereas histidine was less abundant in primary melanoma vs. EM and conversely more abundant in metastatic melanoma vs. EM. UCA is present in the stratum corneum and has been called a "natural sunscreen" due to its photoprotective properties [62]. UCA is synthesized from histidine by the enzyme histidine ammonia lyase [63]. Changes in the expression of histidine ammonia lyase may be related to a transition from primary melanoma to metastatic melanoma.

Our exploratory MSEA pathway-based analysis, which drew on a larger number of physiological pathways and was not limited to KEGG pathways alone, yielded comparable influential network nodes when we compared primary melanoma to EM and metastatic melanoma to EM. L-glutamic acid and pyruvic acid were influential in distinguishing networks of overabundant metabolites in both primary melanoma and metastatic melanoma vs. EM, whereas D-glucose and D-galactose were influential in distinguishing networks of less abundant metabolites in both primary melanoma and metastatic melanoma vs. EM. Glutamic acid influences more metabolic reactions than any other amino acid and serves as a key source of glucose. Glutamic acid has exhibited anti-cancer properties when conjugated with current pharmaceutical treatments (*e.g.* paclitaxel, cisplatin) [64]. In a recent study aimed at differentiating slow and fast proliferative states among melanoma cell lines compared to control media, glutamic acid was observed to play a supportive role in significantly accelerating proliferation, migration, and invasiveness among early stage melanoma cells, but not among metastatic melanoma cells [65].

The overabundance of pyruvic acid we observed among melanoma cells compared to EM is consistent with recent research that suggests it may accumulate in the cytosol as a result of sustained *ERK1/2* activation in melanoma, leading to downregulation of pyruvate dehydrogenase and promotion of lactic fermentation—a hallmark of cancer [66].

The relatively diminished abundance of glucose and galactose among melanoma cells vs. EM is not surprising; melanoma cells are known to metabolize a large proportion of glucose into lactate under both normoxic and hypoxic conditions via the Warburg effect, and galactose is a precursor to glucose production by conversion to glucose 6-phosphate.

## Conclusions

In summary, compounds in the ascorbate and aldarate metabolic pathway distinguish the metabolomes of both primary and metastatic melanoma from that of EM. The tryptophan and propanoate metabolic pathways distinguished primary melanoma from EM, but not metastatic melanoma from EM, suggesting that these pathways may be important to the initiation or

maintenance of the carcinogenic process for melanoma. In contrast, pyrimidine and histidine metabolic pathways could distinguish metastatic melanoma from EM, but not primary melanoma from EM. This finding indicates that these pathways may be important to the progression of melanoma rather than its initiation, and that the proteins or metabolites in these pathways may play a role as potential therapeutic targets to inhibit metastasis.

## Supporting information

**S1 Fig. Study participant and tissue eligibility.** Flow chart illustrating the number of eligible study participants contributing eligible tissue to the study according to matched and unmatched analysis.
(TIF)

**S2 Fig. Number of mass spectrometry peaks identified among eligible tissue specimens.** Flow chart illustrating the number of metabolites (MS peaks) identified among eligible tissue specimens and the final number of metabolites (MS peaks) considered for matched and unmatched analyses after batch correction and quality control measures.
(TIF)

**S3 Fig. Noteworthy differentially abundant metabolites between primary melanoma and EM.** Bar charts reporting noteworthy differentially abundant metabolites between primary melanoma and matched EM.
(PDF)

**S4 Fig. All differentially abundant metabolites between primary melanoma and EM.** Bar charts reporting all differentially abundant metabolites between primary melanoma and matched EM.
(PDF)

**S5 Fig. Noteworthy differentially abundant metabolites between metastatic melanoma and EM.** Bar charts reporting noteworthy differentially abundant metabolites between metastatic melanoma and EM.
(PDF)

**S6 Fig. All differentially abundant metabolites between metastatic melanoma and EM.** Bar charts reporting all differentially abundant metabolites between metastatic melanoma and EM.
(PDF)

**S7 Fig. Metabolite abundance profiles do not differ significantly between primary melanoma and metastatic melanoma tissues.** Heatmap of all metabolites with tissue type (primary melanoma: Red or metastatic melanoma: Black) noted in the top color bar.
(TIF)

**S8 Fig. Metabolite set enrichment analysis results of overabundant metabolites in primary melanoma vs. EM.** Network of metabolite-metabolite interactions is shown among metabolites that were significantly overabundant in primary melanoma vs. EM.
(TIFF)

**S9 Fig. Metabolite set enrichment analysis results of metabolites that were less abundant in primary melanoma vs. EM.** Network of metabolite-metabolite interactions is shown among metabolites that were significantly less abundant in primary melanoma vs. EM.
(TIFF)

**S10 Fig. Metabolite set enrichment analysis results of overabundant metabolites in metastatic melanoma vs. EM.** Network of metabolite-metabolite interactions is shown among metabolites that were significantly overabundant in metastatic melanoma vs. EM. (TIFF)

**S11 Fig. Metabolite set enrichment analysis results of metabolites that were less abundant in metastatic melanoma vs. EM.** Network of metabolite-metabolite interactions is shown among metabolites that were significantly less abundant in metastatic melanoma vs. EM. (TIFF)

## Acknowledgments

Support for this work was also contributed by Moffitt Core facilities, including the Collaborative Data Services Core, the Biostatistics and Bioinformatics Shared Resource, the Tissue Core, and the Metabolomics Core at Moffitt.

## Author Contributions

**Conceptualization:** Nicholas J. Taylor, Peter A. Kanetsky.

**Data curation:** Eric A. Welsh, Timothy J. Garrett, Chris Beecher, Jane L. Messina.

**Formal analysis:** Irina Gaynanova, Steven A. Eschrich, Eric A. Welsh.

**Funding acquisition:** Nicholas J. Taylor, Peter A. Kanetsky.

**Investigation:** Nicholas J. Taylor, Irina Gaynanova, Eric A. Welsh, Timothy J. Garrett, Chris Beecher, Ritin Sharma, John M. Koomen, Keiran S. M. Smalley, Jane L. Messina, Peter A. Kanetsky.

**Methodology:** Nicholas J. Taylor, Irina Gaynanova, Steven A. Eschrich, Eric A. Welsh, Timothy J. Garrett, Chris Beecher, John M. Koomen, Jane L. Messina, Peter A. Kanetsky.

**Project administration:** Nicholas J. Taylor, Peter A. Kanetsky.

**Resources:** Timothy J. Garrett, Chris Beecher, Ritin Sharma, John M. Koomen, Keiran S. M. Smalley, Peter A. Kanetsky.

**Software:** Irina Gaynanova, Steven A. Eschrich, Eric A. Welsh.

**Supervision:** Nicholas J. Taylor, Timothy J. Garrett, Chris Beecher, John M. Koomen, Jane L. Messina, Peter A. Kanetsky.

**Validation:** Irina Gaynanova, Eric A. Welsh, Timothy J. Garrett, Chris Beecher.

**Visualization:** Irina Gaynanova, Steven A. Eschrich, Eric A. Welsh, Timothy J. Garrett, Chris Beecher.

**Writing – original draft:** Nicholas J. Taylor, Irina Gaynanova, Peter A. Kanetsky.

**Writing – review & editing:** Nicholas J. Taylor, Irina Gaynanova, Steven A. Eschrich, Eric A. Welsh, Timothy J. Garrett, Chris Beecher, Ritin Sharma, John M. Koomen, Keiran S. M. Smalley, Jane L. Messina, Peter A. Kanetsky.

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
