## [Decision Letter · Decision Letter 0]

15 Jun 2020

PONE-D-20-09783

Metabolomics of primary cutaneous melanoma and matched adjacent extratumoral microenvironment

PLOS ONE

Dear Dr. Nicholas Taylor,

Thank you for submitting your manuscript to PLOS ONE. After careful consideration, we feel that it has merit but does not fully meet PLOS ONE’s publication criteria as it currently stands. Therefore, we invite you to submit a revised version of the manuscript that addresses the points raised during the review process.

We look forward to receiving your revised manuscript.

Kind regards,

Ch Ratnasekhar, Ph.D.

Academic Editor

PLOS ONE

Journal Requirements:

2. In the ethics statement in the manuscript and in the online submission form, please provide additional information about the tissue samples used in your retrospective study, including: 

a) the date range (month and year) during which patients' medical records were accessed and

b) the date range (month and year) during which patients whose medical records were selected for this study sought treatment.

If patients provided informed written consent to have data from their tissue samples used in research, please include this information.

3. Our staff editors have determined that your manuscript is likely within the scope of our Cancer Metastasis Call for Papers. This editorial initiative is headed by a team of Guest Editors for PLOS ONE: Joe Ramos (University of Hawai'i), Shengyu Yang (Penn State University), Helen Fillmore (University of Portsmouth) and Tobias Zech (University of Liverpool). The Collection will encompass a diverse range of research articles about metastasis, including mechanisms of cell motility, invasion and the tumor microenvironment, as well as advances in the development of anti-metastatic therapies.  Additional information can be found on our announcement page: https://collections.plos.org/s/cancer-metastasis

If you would like your manuscript to be considered for this collection, please let us know in your cover letter and we will ensure that your paper is treated as if you were responding to this call. 

Please note that being considered for the Collection does not require additional peer review beyond the journal’s standard process and will not delay the publication of your manuscript if it is accepted by PLOS ONE.

If you would prefer to remove your manuscript from collection consideration, please specify this in the cover letter.

'I have read the journal's policy and the authors of this manuscript have the following competing interests: Chris Beecher is the Founder and Chief Science Officer of IROA Technologies.  Timothy J. Garrett is a member of the Scientific Advisory Board of IROA Technologies.  IROA® Long Term Reference Standard and Internal Standard were donated by IROA® Technologies.  Timothy J Garrett's laboratory received no funding from IROA® Technologies.'

We note that one or more of the authors are employed by a commercial company: IROA Technologies

Reviewers' comments:

Reviewer's Responses to Questions

**Comments to the Author**

1. Is the manuscript technically sound, and do the data support the conclusions?

Reviewer #1: Yes

Reviewer #2: Yes

Reviewer #3: Partly

2. Has the statistical analysis been performed appropriately and rigorously? 

Reviewer #1: Yes

Reviewer #2: Yes

Reviewer #3: Yes

3. Have the authors made all data underlying the findings in their manuscript fully available?

Reviewer #1: Yes

Reviewer #2: Yes

Reviewer #3: Yes

4. Is the manuscript presented in an intelligible fashion and written in standard English?

Reviewer #1: Yes

Reviewer #2: Yes

Reviewer #3: No

5. Review Comments to the Author

Reviewer #1: Metabolomics-based strategies have become an integral part of present clinical research studies and has been instrumental in cancer precision medicine. This manuscript entitling ‘‘Metabolomics of primary cutaneous melanoma and matched adjacent extratumoral microenvironment’’ by Taylor et al, is an interesting study and authors have collected a unique dataset using an ultra-high-performance liquid chromatography. Overall, this manuscript is well written. However, in my opinion the article has some minor concerns regarding data analyses, and I feel this dataset has not been utilized to its full extent. Furthermore, I would prefer more in-depth functional analyses of the data.

1. Authors did not show any functional analysis of the data. For this, authors can utilize PROFILE clustering to order the metabolites based on their mutual proximity in the metabolic network. This analysis will provide useful insights in the interpretation of metabolomic changes in primary and metastatic melanoma and EM.

2. Authors should create a metabolic map based on the differential expression of metabolites in the primary tumor and EM by using a molecular interaction software like Cytoscape. This metabolic map can be used for predicting predominant metabolic processes in melanoma and will help in building strategies for therapeutic targeting of metabolic pathways in cancer cells.

Reviewer #2: The current manuscript describing the LC/MS base metabolomics analysis of melanoma and its matched adjacent extratumoral microenvironment tackles an especially interesting subject. The work done is substantial well designed and executed whereas the statistical treatment of the results was in my opinion accurate and also well designed. The IROA technology used has also led to increased confidence concerning the identification of features. Finally, the authors explained their findings convincingly, rendering the work o significant merit for the readers.

A couple of minor points should be taken care of.

line 157 please specify if the external calibration refers to mass

line 174 using pooled QC samples is a frequently used procedure in metabolomics analyses. Did the authors consider to employ this methodology in order to control the intra batch variation as well? Please comment

Table 1 Does the second digit in retention time hold a meaning? Please comment

Please also add the statistics of multivariate analyses i.e. at least the R2 and Q2, to indicate their statistical significance.

Reviewer #3: Comments

Article no. Ms. Ref. No. PONE-D-20-09783, Research Article entitled ‘Metabolomics of primary cutaneous melanoma and matched adjacent extratumoral microenvironment’.

In the present study, authors have analysed the metabolomics of primary cutaneous melanoma and matched adjacent extratumoral microenvironment using ultra-high-performance liquid chromatography-mass spectrometry. There are few novel findings and may also hold promising translational relevance to develop the novel biomarker for melanoma patients. However, I do notice many important concerns that should be addressed to further enhance the manuscript quality:

Comment 1: First of all, authors need to provide high resolution images for all figures as it’s very hard to figure out anything out of any heatmaps.

Comment 2: Authors really need to explain their results in different subheadings and need to cite figure numbers in text rather than in headings.

Comment 3: It’s better to provide the box plot or bar diagram of potential differentially altered metabolites with their respective p-values in addition to heatmap.

Comment 4: Authors need to analyse and correlate the altered level of their identified metabolites with other pathophysiological characteristics such as TNM staging and survival if possible.

Comment 5: Authors need to provide demographic and clinical-pathological details of all the clinical bio specimens collected in their study.

6. PLOS authors have the option to publish the peer review history of their article (what does this mean?). If published, this will include your full peer review and any attached files.

Reviewer #1: Yes: Manish Charan

Reviewer #2: No

Reviewer #3: No

---

## [Author Response · Author response to Decision Letter 0]

14 Sep 2020

Reviewer #1

1) “Authors did not show any functional analysis of the data. For this, authors can utilize PROFILE clustering to order the metabolites based on their mutual proximity in the metabolic network. This analysis will provide useful insights in the interpretation of metabolomic changes in primary and metastatic melanoma and EM.”

“Authors should create a metabolic map based on the differential expression of metabolites in the primary tumor and EM by using a molecular interaction software like Cytoscape. This metabolic map can be used for predicting predominant metabolic processes in melanoma and will help in building strategies for therapeutic targeting of metabolic pathways in cancer cells.”

We have given these suggestions careful consideration. Our original functional analysis was a pathway-based approach via Metaboanalyst and Mummichog focused on mapping compounds to known KEGG pathways in human beings. Our intention was, and remains, to highlight potentially meaningful differences in biological pathways between tissue types. We retain this approach as our primary analysis in our revised manuscript.

However, in order to fully address the Reviewer’s comments, and to further explore the functional characteristics of the metabolite differences between tissue types, we performed a metabolite set enrichment analysis (MSEA) within Metaboanalyst 4.0 as a secondary analysis. Again, we considered metabolites with statistically significantly higher abundance (or lower abundance) in primary melanoma (or metastatic melanoma) compared to EM. For each condition, the MSEA of pathway-associated metabolite sets—based on the small molecule pathway database (SMPDB—smpdb.ca)—was considered. A network of metabolite-metabolite interactions was plotted for each condition, with nodes representing metabolites and edges indicating a relationship (from STITCH—an interaction network database for small molecules and proteins) between metabolites. STITCH incorporates several databases (including KEGG) when producing the “interactome” of metabolites (e.g. Protein Data Bank, Reactome, NCI-Nature Pathway Interaction Database, DrugBank, and MATADOR) We highlighted specific KEGG pathways within the network for comparison to our main analysis.

We have made appropriate additions to the Methods and Results sections describing these secondary analyses, and we have highlighted the relevant text in the revised manuscript. We also present the results of these exploratory analyses in supplemental material (Supplemental Figures 8-11), and add some discussion as highlighted in the manuscript.

Reviewer #2

1) “line 157 please specify if the external calibration refers to mass.”

External calibration refers to both mass calibration and mass resolution calibration; we have added this clarification to the methods section of the revised manuscript. 

2) “line 174 using pooled QC samples is a frequently used procedure in metabolomics analyses. Did the authors consider to employ this methodology in order to control the intra batch variation as well? Please comment.”

We did not use pooled QC samples in this study. The melanoma tissue samples are a scarce resource (especially the primary melanoma samples) and were generally low in mass; the decision was made to preserve tissue for multiple omics studies and other assays rather than using a pool for batch-to-batch quality control. We used a pooled red cross plasma sample to assure instrument performance in both retention time and mass accuracy. Our process of detecting and mitigating inter-batch bias is presented in detail in the manuscript from lines 175-182, as well as Figure 1. 

3) “Table 1 Does the second digit in retention time hold a meaning? Please comment.”

We thank the reviewer for pointing out this oversight. We have removed the unnecessary digits from all reports of retention times in the manuscript.

4) “Please also add the statistics of multivariate analyses i.e. at least the R2 and Q2, to indicate their statistical significance.”

The Authors are confused by the Reviewer’s comment; we did not perform multivariate analyses of these high dimensional data.

Reviewer #3

1) “First of all, authors need to provide high resolution images for all figures as it’s very hard to figure out anything out of any heatmaps.”

We previously (and again) provided the journal with high resolution images conforming to PLoS ONE guidelines and requirements. Please note that images/figures contained in the manuscript pdf file appear in lower resolution; however, the url links contained within the pdf file allow for images/figures to be viewed externally at full resolution.

2) “Authors really need to explain their results in different subheadings and need to cite figure numbers in text rather than in headings.”

We have added subheadings to the results section per the Reviewer’s suggestion. The formatting of figure citations in the text of the manuscript was made to adhere to PLoS ONE requirements.

3) “It’s better to provide the box plot or bar diagram of potential differentially altered metabolites with their respective p-values in addition to heatmap.”

We now provide bar diagrams illustrating a) significantly differentially abundant metabolites between primary melanoma and paired EM tissues which we highlight as being identified and named in both positive and negative ion modes, as well as two identified and named metabolites that are differentially abundant between primary melanoma and paired EM in separate ion modes (Supplemental Figure 3)—from results section lines 237-239; and, b) all significantly differentially abundant metabolites between primary melanoma and paired EM (Supplemental Figure 4).

Further, we now include box plots illustrating differentially abundant metabolites between EM and unmatched metastatic melanoma tissues. Supplemental Figure 5 illustrates results described in the Metastatic melanoma vs. EM section in lines 255-261; Supplemental Figure 6 illustrates all significantly differentially abundant metabolites between EM and unmatched metastatic melanoma. 

4) “Authors need to analyse and correlate the altered level of their identified metabolites with other pathophysiological characteristics such as TNM staging and survival if possible.”

We appreciate the Reviewer’s suggestion, and we have previously considered this approach. We decided against including results from such analyses in the manuscript for important reasons: 1) Due to the scarcity of primary melanoma tissue, our study sample essentially represents a convenience sample. This limits our ability to select optimal tissue from those patients who developed multiple primary melanomas; moreover, we were limited in obtaining data indicating whether or not a patient had multiple primary melanomas. This limitation hinders our ability to make reasonable inferences from survival analyses; 2) Our sample size is relatively small in comparison to the number of metabolites identified, which makes external validation highly variable and power low.

Nevertheless, we convey results from 3 different analyses of the data with respect to survival below.

1) We performed a Cox proportional hazard model on the raw difference between tumor and EM (i.e. tumor minus EM) and the fold difference (i.e. tumor divided by EM) for each of the 824 statistically significant metabolites identified. The resulting p-values are all non-significant after adjustment for FDR, indicating they are either not important for survival (unlikely) or we have insufficient power to detect their effects (likely).

2) We binned 33 subjects into two clusters based on the profiles of the 824 statistically significant metabolites identified and compared the resulting Kaplan-Meier survival curves. There is no significant survival separation between the two clusters (again, likely due to insufficient power).

3) We performed a cross-validation with penalized Cox regression model on the 824 differentially abundant metabolites (meaning ridge regression so that all metabolites are used in the prediction). The 33 samples were split into 10 folds (100 splits total, each has 10 folds); the best model was fit on 9 folds and the risk evaluated on the remaining one. The measure of model prediction is Harrel’s concordance (https://statisticaloddsandends.wordpress.com/2019/10/26/what-is-harrells-c-index/), with values of 0.5 equating to a completely random guess and 1.0 being maximum prediction. Results from this approach were highly variable across 100 splits and range from 0.3 to 0.8. On average, the best model with 824 metabolites on EM tissue gave better prediction than the best model with 824 metabolites on tumor tissue (concordance means 0.58 vs. 0.52), though both means were similar and close to 0.5, indicating metabolite abundances in EM and primary tissues are not predictive of survival in our dataset (or likely we have insufficient power to detect their effects).

Lastly, the Reviewer suggested analyses of associations between metabolites and pathophysiologic characteristics. Unfortunately, we were unable to reliably perform these analyses due to the limited data we had on these factors.

5) “Authors need to provide demographic and clinical-pathological details of all the clinical bio specimens collected in their study.”

We have added a new Table 1, which now includes descriptive summaries of sex, vital status, age at diagnosis, histology, TNM staging, anatomic site of primary, and anatomic site of metastasis.

---

## [Editor Report · Decision Letter 1]

5 Oct 2020

Metabolomics of primary cutaneous melanoma and matched adjacent extratumoral microenvironment

PONE-D-20-09783R1

Dear Dr. Nicholas,

We’re pleased to inform you that your manuscript has been judged scientifically suitable for publication and will be formally accepted for publication once it meets all outstanding technical requirements.

Kind regards,

Ch Ratnasekhar, Ph.D.

Academic Editor

PLOS ONE
---

## [Editor Report · Acceptance letter]

16 Oct 2020

PONE-D-20-09783R1 

Metabolomics of primary cutaneous melanoma and matched adjacent extratumoral microenvironment 

Dear Dr. Taylor:

I'm pleased to inform you that your manuscript has been deemed suitable for publication in PLOS ONE. Congratulations! Your manuscript is now with our production department. 

Kind regards, 

on behalf of

Dr. Ch Ratnasekhar 

Academic Editor

PLOS ONE